# A Novel Framework for Analysis of the Shared Genetic Background of Correlated Traits

**DOI:** 10.3390/genes13101694

**Published:** 2022-09-21

**Authors:** Gulnara R. Svishcheva, Evgeny S. Tiys, Elizaveta E. Elgaeva, Sofia G. Feoktistova, Paul R. H. J. Timmers, Sodbo Zh. Sharapov, Tatiana I. Axenovich, Yakov A. Tsepilov

**Affiliations:** 1Institute of Cytology and Genetics, Siberian Branch of the Russian Academy of Sciences, 630090 Novosibirsk, Russia; 2Vavilov Institute of General Genetics, Russian Academy of Sciences, 117971 Moscow, Russia; 3Novosibirsk State University, 630090 Novosibirsk, Russia; 4MRC Human Genetics Unit, MRC Institute of Genetics and Cancer, University of Edinburgh, Edinburgh EH8 9YL, UK; 5Centre for Global Health Research, Usher Institute, University of Edinburgh, Edinburgh EH8 9YL, UK

**Keywords:** GWAS, shared genetic component, linear combination of traits, shared heritability, proportion of heritability explained by SGF

## Abstract

We propose a novel effective framework for the analysis of the shared genetic background for a set of genetically correlated traits using SNP-level GWAS summary statistics. This framework called SHAHER is based on the construction of a linear combination of traits by maximizing the proportion of its genetic variance explained by the shared genetic factors. SHAHER requires only full GWAS summary statistics and matrices of genetic and phenotypic correlations between traits as inputs. Our framework allows both shared and unshared genetic factors to be effectively analyzed. We tested our framework using simulation studies, compared it with previous developments, and assessed its performance using three real datasets: anthropometric traits, psychiatric conditions and lipid concentrations. SHAHER is versatile and applicable to summary statistics from GWASs with arbitrary sample sizes and sample overlaps, allows for the incorporation of different GWAS models (Cox, linear and logistic), and is computationally fast.

## 1. Introduction

There is a growing interest in studying the shared genetic background between genetically correlated traits [1,2,3,4,5] (see, for example, the number of PubMed search results by year for keywords related to “shared genetic background”). Studying the shared genetics between traits can help with the discovery of pleiotropic interactions, common genes and pathways, and identify genetic effects that are specific for each trait. 

The problem of the decomposition of the variance of several traits into the shared/unshared genetic and environment components were first formulated by S. Wright in 1921 [6]. There are widely used classic twin designs to address this problem. They are based on structural equation modelling; in particular, multivariate pathway models assuming the existence of the genetic influences common for all traits and specific for each trait [7]. These designs are implemented only for the variance decomposition, but not for the identification of the genetic factors that determine these genetic effects. 

There are several terms for these common and specific genetic impacts. We will call them the “shared genetic impact” (SGI) and “unshared genetic impacts” (UGI). The genetic factors that determine these impacts will be called “shared genetic factors” (SGF) and “unshared genetic factors” (UGF), respectively. The heritability of each trait explained by SGF and UGF will be called “shared heritability” and “unshared heritability”, respectively. Note that the term “unshared” for every trait means the rest other than the “shared”. For any incomplete set of traits, UGFs can be partially overlapping.

The application of different methods of multivariate analysis in genome-wide association studies (GWAS) allows the problem of SGF and UGF identification to be partially solved [8,9,10,11,12,13]. The multivariate methods involve complicated genetic or/and phenotypic correlation structures of traits in the analysis. In most cases, this increases the power of detection of the loci associated with several traits due to pleiotropic effects. If the detected locus has a pleotropic effect on all studied traits, the locus could potentially be attributed to SGF, and if not, to UGF. However, a pleiotropic effect of the locus on all studied traits is necessary but insufficient for inclusion of this locus in SGF (at least effects should be also collinear between traits; see the model description below). Moreover, this approach of SGF identification assumes a manual classification of loci, which prevents the use of more sophisticated modern in-silico approaches for genetic analysis, for example, the ones that rely on GWAS summary statistics [14]. To our knowledge, there is no specific method that could be good for both variance component decomposition and identification of SGF and UGF.

We had previously developed a method for obtaining genetically independent phenotypes (GIPs) [2]. This method is based on the calculation of the principal components using genetic rather than phenotypic correlations. We applied this method to genetically correlated pain phenotypes and aging related phenotypes and showed that the first GIP component, GIP1, that explains the largest proportion of the genetic variance probably could be interpreted as SGI [2,15]. This makes GIP promising for the identification of loci attributed to SGF. However, this method was not designed specifically for SGI analysis. In addition, no specific experiments have been performed to validate the approach or to estimate its statistical properties.

Here, we present a novel general framework for the estimation of shared and unshared heritability and identification of the shared and unshared genetic factors using the summary statistics of original traits. The essence of our approach is to find the optimum linear combination of traits which has the maximum proportion of its genetic variance explained by the SGF. We validated our framework using simulation studies under different scenarios by comparing it with the developed GIP approach, and assessed its performance using three real datasets: anthropometric indices, psychiatric disorders and conditions, and lipid concentrations.

## 2. Materials and Methods

### 2.1. Shared Heredity Model

We adopted a commonly used multivariate pathway model [7] in terms of SGF and UGF. We call it the “shared heredity model”. For simplicity, we consider SGF and UGF as biallelic SNPs and consider a sample of *N* unrelated individuals measured for *K* traits and genotyped for *M* SNPs. For a standardized normal trait, *y* (*N ×* 1), the traditional polygenic (null) model takes the form: *y* = *Gβ* + *ε*, where *G* is an (*N × M*) matrix of standardized genotypes; *β* (*M ×* 1) and *ε* (*N* × 1) are genetic and non-genetic random effects, respectively; *β* ~ *N*(**0**, *h*^2^*I_M_*) and *ε* ~ *N*(**0**, (1 − *h*^2^)*I_N_*), where **0** is a null mean vector, *h*^2^ is the trait heritability, and *I* is an identity matrix of the given dimension. For unrelated individuals, we expect *y* ~ *N*(**0**, *I_N_*). 

We propose to divide *M* SNPs into two non-overlapping SNP sets with sizes *M*_0_ and *M*_1_ (*M*_0_ + *M*_1_ = *M*). The set of *M*_0_ SNPs called “SGF” includes only those SNPs whose effects on all traits are collinear. The set of *M*_1_ SNPs consists of the other SNPs which do not have shared joint influence on all traits at once, this set being called “UGF”. In accordance with *M, G* is divided into two matrices, *G*_0_ (*N* × *M*_0_) and *G*_1_ (*N* × *M*_1_). To decompose every trait into components explained by SGF and UGF, we rewrote the traditional polygenic model in terms of *G*_0_ and *G*_1_
(1)$$y_{i}=\underset{due\;to\;SGF}{\underbrace{G_{0}b_{0_{i}}}}+\underset{\begin{array}{c} due\;to\;UGF\; \end{array}}{\underbrace{G_{1}b_{1_{i}}}}+\varepsilon_{i}.$$

Here, the first and second terms are genetic components explained by SGF and UGF, respectively, which are assumed to be independent. In the first term, $b_{0_{i}}$ is an (*M*_0_ × 1) vector of non-zero SGF effects, which can be presented as $\beta_{0}w_{i}\sqrt{h_{i}^{2}}$, where *β*_0_ is an (*M*_0_ × 1) non-zero vector that is the same for all traits, $\beta_{0}\sim N\left(0,I_{M_{0}}\right)$, and $w_{i}^{2}h_{i}^{2}$ is the heritability of the *i*-th trait explained by SGF. Here *w_i_* is a non-zero trait-specific multiplier: *w_i_*^2^ denotes the proportion of *h_i_*^2^ explained by SGF; and the value of *w_i_* can be positive and negative, indicating the direction of the SGF effect on the *i*-th trait. $G_{0}\beta_{0}$ is the so-called shared genetic impact or SGI. In the second term of Model (1), $b_{1}{}_{i}$ is an (*M*_1_ × 1) vector of UGF effects, which can be presented as $b_{1i}=\beta_{1i}\sqrt{\left(1-w_{i}^{2}\right)h_{i}^{2}}$, $\beta_{1i}\sim N\left(0,I_{M_{1}}\right)$. In contrast to *β*_0_, $\beta_{1}{}_{i}$ are different for different traits, and they are not collinear. For illustrative purposes, we rewrote Equation (1) as:(2)$$y_{i}=\underset{due\;to\;SGF}{\underbrace{\underset{SGI}{\underbrace{G_{0}\beta_{0}}}w_{i}\sqrt{h_{i}^{2}}}}+\underset{due\;to\;UGF}{\underbrace{G_{1}\beta_{1_{i}}\sqrt{1-w_{i}^{2}}\sqrt{h_{i}^{2}}}}+\varepsilon_{i}.$$

### 2.2. Overview of the SHAHER Framework

For analyses of SGI and UGI on a set of correlated traits, we propose an effective multi-stage framework named SHAHER (see Figure 1). The concept of the framework is first to partition the genetic basis of each original trait into two components: one shared by all the original traits and one not shared by all the original traits, and then to identify the SNPs that contribute to these genetic components. To do this, we propose to construct new traits: (1) an SGIT as a linear combination of original traits, which has the maximum possible heritability explained by SGF, and (2) UGITs as linear combinations of the original traits, which are obtained by adjusting the original traits for SGIT. This means that the genetic basis of UGITs is predominantly determined by UGF. 

Our framework requires matrices of phenotypic correlations (*U_phen_*) between the original traits, the matrices of genetic correlations (*U_gen_*) between the original traits, the heritabilities of the original traits, and GWAS summary statistics of the original traits as inputs. It is worth noting that *U_phen_*, *U_gen_* and heritabilities could be estimated using GWAS summary statistics of the original traits, for example, by the LD score regression method [16].

SHAHER starts with a preliminary stage, which verifies the presence of SGI in a given set of traits. This is achieved by checking the following requirements for *U_gen_*: it must be positive definite; the absolute values of its elements must be significantly more than a given threshold, and the rank of the correlation matrix derived from *U_gen_* by rounding its elements to extremal correlation values, either −1 or 1, must be equal to one. If the requirements are met, we turn to the basic stages of SHAHER.

*The MaxSH stage*. To determine the *α* and *γ* coefficients for the linear combinations of the original traits to build SGIT and UGITs, we developed the MaxSH method, which is based on the correlation component model given below. This model partitions the phenotypic correlation matrix, *U_phen_*, into environmental and genetic components, *U_env_* and *U_gen_*, respectively, the latter being further subdivided into two components caused by SGF and UGF:(3)$$\begin{array}{c} U_{phen}=\underset{genetic\;component}{\underbrace{\sqrt{H^{2}}U_{gen}\sqrt{H^{2}}}}+\underset{environmental\;component}{\underbrace{\sqrt{I-H^{2}}U_{env}\sqrt{I-H^{2}}}}\\ U_{gen}=\underset{due\;to\;SGF}{\underbrace{W11^{T}W}}+\underset{due\;to\;UGF}{\underbrace{\sqrt{I-W^{2}}U_{unsh}\sqrt{I-W^{2}}}} \end{array}$$

Here *W* is a diagonal matrix, whose *i*-th diagonal element is *w_i_*; *U_unsh_* is a matrix of genetic correlations explained by UGF; *H*^2^ is a diagonal matrix, whose *i*-th diagonal element is *h_i_*^2^, and **1** is a (*k* × 1) vector of units. Using this model, MaxSH solves several tasks. 

First of all, using only the genetic correlation matrix, *U_gen_*, we estimate the proportion of heritability of every trait explained by SGF (*W*). To do this, we minimize the difference between *U_gen_* and the auxiliary matrix *V*. This matrix is built using Formula (2), with the identity matrix used instead of *U_unsh_*. The second task is to determine the *α*-coefficients, which is solved by maximizing the shared heritability of SGIT. This task is analytically solved as
(4)$$a=\frac{U_{phen}^{-\frac{1}{2}}HW1}{\sqrt{1^{T}WHU_{phen}^{-1}HW1}}$$

It requires *U_phen_*, *H*^2^ and *W* as input data. 

After determining the *α*-coefficients and building SGIT, we build a UGIT for every trait using the residual regression equation *UGIT_i_* = *y_i_* − *SGIT***c_i_*, where *c_i_* is the impact of SGIT on the *i*-th original trait, defined as
(5)$$c_{i}=\frac{cov_{gen}\left(y_{i},SGIT\right)}{h_{SGIT}^{2}}.$$

Here *cov_gen_* denotes a genetic covariance. Note that we should use genetic rather than phenotypic covariances, as our goal is to adjust only the genetic components of the original traits.

Since SGIT is the linear combination of the original traits, UGITs are linear combinations of the original traits, as well. The coefficients of these linear combinations called the *γ*-coefficients form the matrix of the *γ*-coefficients $\Gamma =\left(I_{K}-\alpha c^{T}\right)$, where the *i*-th column of *Γ* corresponds to the linear combination coefficients for building the *i*-th UGIT. 

*The sumCOT stage.* This stage is aimed at obtaining GWAS summary statistics for SGIT and UGITs using the previously determined *α* and *γ* coefficients, GWAS summary statistics (Z-scores, allele frequencies and sample sizes for each SNP) for the original traits and the matrix of phenotypic correlations. The method can use Z scores obtained from any regression model and allows for varying sample sizes and sample overlap between traits. This sample overlap is incorporated into the estimation of the matrix of phenotypic correlations. In short, the SNP effects for combined traits are calculated by summing effect estimates from the individual trait GWASs, each multiplied by their corresponding linear coefficient (*α* or *γ*), and standardized by the expected variance. The standard errors of the SNP effect are calculated using variance-covariance arithmetic, taking into account the phenotypic covariance between GWAS results to adjust for the sample overlap. Effective sample sizes are then estimated based on the median Z statistic and allele frequencies by solving Equation (1) in [17]. 

At the final stage, SHAHER checks for the correctness of the output. In particular, we anticipate that UGITs do not have a shared genetic basis. This is verified by applying MaxSH to the matrix of correlations between UGITs. 

To summarize, our framework estimates shared and unshared heritabilities for each of the studied original traits and produces GWAS summary statistics for SGIT and UGITs as outputs. 

The full details and mathematical formulae of SHAHER are in Appendix A.

### 2.3. Simulation Study

Under different scenarios, we designed simulations to assess the performance of MaxSH. We (1) assessed the accuracy of *w* estimates, (2) assessed the proportion of SGIT heritability explained by SGF to the total heritability of SGIT (the *Q*-value), and (3) compared the analytically predicted total and shared heritabilities of SGIT and GIP1 with respect to the loss function. The design of our simulation experiment is shown in Figure 2 To generate the input for the MaxSH and GIP approaches, we used a six-parameter simulation model, in which *K* is the number of traits; *W*_0_^2^ is a (*K* × *K*) diagonal matrix, where the *i*-th diagonal element is *w_i_*^2^ (the proportion of heritability explained by SGF); *s* is the proportion of zeros in the matrix *U_unsh_*; *d*_1_ is the amplitude of the uniform distribution for non-zero values of *U_unsh_* and *d*_2_ is the amplitude of the uniform distribution for *U_env_*; *H*^2^ is the diagonal matrix with diagonal elements equal to the trait heritabilities. The parameter values used are given in Figure 2. 

For each fixed number, *K*, of the original traits and fixed heritability, *h_i_*^2^ (*i* = 1, …, *K*), of each trait, we simulated *U_gen_*. To do this, we separately modelled two of its components caused by SGF and UGF as $W11^{T}W$ and $\sqrt{I-W^{2}}U_{unsh}\sqrt{I-W^{2}}$, respectively (see the “Model” box in Appendix A). Here **1** is a (*K* × 1) vector of units, and *U_unsh_* is a (*K* × *K*) matrix randomly generated using the parameters *s* and *d*_1_ (see Appendix A). We then randomly generated the trait-trait correlation matrix *U_env_* explained by the environmental factors, by giving the parameter *d*_2_ (see Appendix A). Finally, we modeled a matrix of phenotypic correlations by using Model (2) with regard to simulated values *W*_0_.

Using simulation data *U_phen_*, *U_gen_* and *H*^2^, we estimated *W_est_* and calculated its squared relative difference with the simulated values of *W*_0_ (Δ*W*). We revealed a dependence of Δ*W* on the loss function (*Loss*). The *Loss* value characterizes the difference between *U_gen_* and the auxiliary matrix *V*.

We then estimated α in three ways: (1) using MaxSH and *W*_0_, (2) using MaxSH and *W_est_*, and (3) using the GIP method [2]. Based on these estimates, we formed three traits being the linear combinations of the original traits. For these combined traits, we calculated the total heritability and the heritability explained by SGF.

The simulated experiments were repeated 10,000 times for each set of parameters. The model parameters and formulas for all calculated values are shown in Figure 2.

### 2.4. Application to Real Data

#### 2.4.1. Data Sets

We used three publicly available real data sets: anthropometric traits, psychiatric conditions and lipid concentrations, which contain five, four and three traits, respectively. 

The group of anthropometric traits consisted of UK Biobank GWAS summary statistics obtained from the Neale lab (http://www.nealelab.is/blog/2017/7/19/rapid-gwas-of-thousands-of-phenotypes-for-337000-samples-in-the-uk-biobank, accessed on 1 September 2020) for people of European ancestry: BMI (*N* = 336,107), weight (*N* = 336,227), hip (*N* = 336,601), waist circumference (*N* = 336,639) and whole body fat mass (*N* = 330,762).

The second dataset reflecting psychometric traits was constructed from GWAS results provided by the Psychiatric Genomics Consortium (https://www.med.unc.edu/pgc/download-results/, accessed on 1 September 2020) for bipolar disorder, BIP (N cases = 20,352; N controls = 31,358) [18], major depressive disorder, MDD (N cases = 43,204; N controls = 95,680; without UK Biobank and 23andMe data) [19] and schizophrenia, SCZ (N cases = 36,989; N controls = 113,075). Summary statistics for the fourth trait–subjective well-being (*N* = 110,935)–were derived from UK Biobank data from the Neale lab. All psychometric trait GWASs were conducted using samples from white Europeans.

The last dataset corresponding to lipid traits was formed using GWAS data for European participants from the Global Lipid Genetics Consortium (http://csg.sph.umich.edu/willer/public/lipids2013/, accessed on 1 September 2020) for LDL cholesterol (*N* = 173,082), triglycerides (*N* = 177,860), and total cholesterol (*N* = 187,365).

Summary statistics for the three data sets were integrated and quality controlled by the GWAS-MAP platform developed by our group [20]. The GWAS-MAP database contains implemented software for quality control of GWAS results, the estimation of phenotypic correlations, and LD Score regression (LDSC) [16].

We conducted the quality control of all data and unified them within the GWAS-MAP platform [20]. We filtered all summary statistics by minor allele frequencies ≥ 0.01. Additionally, we filtered GWAS results for BIP by imputation qualities ≥ 0.9. We did not apply this filter to the other traits due to the absence of imputation quality in summary statistics data. Finally, using GWAS-MAP, we performed a correction for genomic control for all traits (including the original traits, SGIT and UGITs) with an LDSC intercept greater than 1 [16]. Thus, we corrected all traits from the psychometric dataset apart from MDD, all original anthropometric traits and their SGIT and lipid SGIT, as their LDSC intercept exceeded 1 (see Appendix A). Moreover, all SNPs with a *p*-value equal to 0 were excluded from analysis.

#### 2.4.2. Genetic Analysis

Pairwise phenotypic correlations between traits were computed using the GWAS-MAP platform described above. The used method is based on correlations between insignificant *z*-statistics for independent SNPs as previously described in [9]. SNP-based heritability and genetic correlation coefficients were estimated using the LD Score regression software [16] embedded in the GWAS-MAP platform. The significance threshold for genetic correlations was set at 4.5 × 10^−4^ (0.05/112, where 112 is the number of pairwise combinations between all original traits, their SGIT and UGITs in each dataset–between 11, nine and seven traits for anthropometry, psychometric and lipid traits, respectively).

The SHAHER analysis included checking if there was an SGI or not, the application of MaxSH, and conducting SGIT and UGIT GWASs. The threshold for confirming the existence of an SGI at the first stage was empirically set to 0.2.

For each dataset, we visualized the full genetic correlation matrices using the *corrplot*() function from the corrplot R package (v.0.84) [21]. We also placed the SNP-based heritability estimates on the diagonal and crossed out non-significant values.

Finally, we compared the GWAS results obtained for SGIT by MaxSH and GIP (the principal component analysis on the matrix of genetic covariances) [2]. 

#### 2.4.3. Gene set and Tissue/Cell Type Enrichment Analyses

We performed a gene set enrichment analysis and a tissue/cell type enrichment analysis combined with a gene prioritization using the Data-driven Expression Prioritized Integration for Complex Traits (DEPICT) tool v.1.1, release 194 [22]. We selected genome-wide significant SNPs (*p*-value < 5 × 10^−8^) from summary statistics before the genomic control and applied the DEPICT software with default parameters (https://data.broadinstitute.org/mpg/depict/, accessed on 1 September 2020). The MHC region was excluded from analysis. 

Next, for the gene set enrichment results, we calculated the number of significant enriched gene sets (FDR < 5%) and constructed an overlapping matrix in which each cell represents the number of overlapping gene sets for each pair of traits. For each pair of traits, we scaled the number of overlapping gene sets by the minimum number of significant gene sets for this pair of traits. The resulting matrix was visualized using the corrplot R-package, as descried above. 

#### 2.4.4. The Number of Original Traits Associated with SGIT Loci

We performed a clumping procedure to search for loci associated with each of the original traits, SGIT and UGITs at a genome-wide significance level of 5 × 10^−8^. The associated locus was defined as a genomic region spanning 500 kb in either direction of the lead SNP. Those loci that were significantly associated with SGIT, but not with the original traits, were assumed to be new loci. 

We expected that the loci associated with all the original traits used to obtain SGIT were likely to be SGF. To test this expectation, for each dataset we selected all independent loci that were significantly associated with at least one of the original traits and calculated the number of the original traits significantly associated with these loci. For the original anthropometric and lipid traits, we empirically set the significance threshold at *p*-value = 1 × 10^−5^. For the psychometric traits, it was set at 1 × 10^−3^. We then analyzed the SGIT *p*-values for the selected loci and constructed boxplots of −log_10_ for them with regard to the number of the original traits significantly associated with these loci.

## 3. Results

### 3.1. Simulation Study

To assess the MaxSH performance, we conducted simulation studies. We (1) assessed the accuracy of *w* estimates (using ΔW metrics estimated as ${\left(\frac{w_{0}-w_{est}}{w_{o}}\right)}^{2}$, where *w*_0_ and *w_est_* are modeled and estimated *w*, respectively) with respect to the loss function given in Figure 2; (2) assessed the proportion of the shared heritability to the total heritability of SGIT (the *Q*-value) with respect to the loss function; and (3) compared the analytically predicted total/shared heritabilities of two traits: SGIT and the first component, GIP1, obtained by the GIP method [2]. The *Q*-value can be interpreted as the specificity metrics of SGIT: the closer the *Q*-value to 1, the lower the share of unshared heritability in the total heritability of SGIT. The simulation scenarios were based on six varying parameters that describe the properties of the genetic and phenotypic correlation matrices. Under each scenario, we considered two situations where all traits have the same *w*^2^ and different *w*^2^s. To distinguish between these situations, we will hereinafter write either “*w*^2^” or “different *w*^2^s”. In total, we performed 10,000 iterations of simulations for each of 288 scenarios. 

The full results are presented in Appendix A. For all scenarios, there are few general patterns: (1) the higher simulated *w* values, the higher the accuracy of the *w* estimates, (2) the accuracy of the *w* estimates and the *Q*-value increase with an increasing in the number, *K*, of traits, (3) for all scenarios with *w*^2^ > 0.8, ΔW was very low (<0.025) and the *Q*-value was more than 90%.

For all scenarios with three traits, the accuracy of the *w* estimates was generally low: ΔW was not higher than 0.7 for scenarios with *w*^2^ = 0.2 and 0.3, although at *w*^2^ equal to or higher than 0.4, ΔW was less than 0.2. The *Q*-value was higher than 60% for almost all scenarios with *w*^2^ ≥ 0.4. 

For the scenarios with four and five traits, the accuracy of *w* estimates was higher: ΔW < 0.15 for *w*^2^ ≥ 0.4 and ΔW < 0.05 for *w*^2^ ≥ 0.5. For the scenarios with *w*^2^ ≥ 0.5, the *Q*-value was more than 70% for four traits and more than 80% for five traits. 

The selected results of comparison of the shared heritability of SGIT and the shared heritability of GIP1 for scenarios with *s* = 0.3 are presented in Figure 3. The results with *s* = 0.8 were similar to those with *s* = 0.3. They are presented in Appendix A. For three traits in almost all cases, the shared heritabilities of SGIT were higher than the corresponding heritabilities of GIP1, except for the scenarios with *h*^2^ = 0.8. For four and five traits, the shared heritabilities of SGIT were higher than the corresponding heritabilities of GIP1 under all scenarios, except for the scenarios with *h*^2^ = 0.8. In the scenarios with *h*^2^ = 0.8, the shared heritabilities of SGIT were higher than those of the GIP1 at *w*^2^ ≥ 0.5. The patterns of the total heritabilities for all scenarios reproduced the corresponding patterns of the shared heritabilities (Appendix A).

In summary, the performance of MaxSH was suitable at *w*^2^ ≥ 0.5 and when the number of traits was higher than or equal to four. In the case of small *w* or three traits, the results of MaxSH should be interpreted with caution. 

### 3.2. Real Data Assessment

We applied SHAHER to three datasets: anthropometric (five traits), psychometric (four traits) and lipid traits (three traits). We should note that the performance of SHAHER applied to three traits is limited (see simulation results), yet still passable, although the results should be interpreted with caution. The number of identified loci for each trait for each data set is given in Table 1. We present SHAHER results for anthropometric traits in the main text as an example. The full results for the psychometric and lipid traits are presented in Appendix A.

At the first step, we confirmed that SGI exists for five traits. At the second step, we determined the *α* and *γ* coefficients and their CI (see Appendix A). At the third step, we applied sumCOT and obtained GWAS results for SGIT and UGITs (see Appendix A for heritability estimates and LD score regression intercepts). SHAHER results are presented in Figure 4.

Figure 4A demonstrates genetic correlations between all pairs of the original anthropometric traits, SGIT and UGITs. All the original traits were positively correlated with r > 0.82. We did not observe any significant genetic correlation between SGIT and UGITs. Moreover, we did not observe additional SGI among UGITs, which was expected. The heritabilities of UGITs varied from 0.07 to 0.14. 

We revealed a dependence of the SGIT *p*-value from the number of the original traits significantly associated with the locus (Figure 4B). It clearly shows that the loci associated with all the original traits have lower SGIT *p*-values than the other loci. 

Joint clumping of 11 traits (five original traits, five UGITs and SGIT) resulted in 820 genome-wide significantly associated loci (*p*-value < 5 × 10^−8^, Appendix A). If a locus was not significantly associated with any of the original traits, it was considered new. SGIT was significantly associated with 296 SNPs. We detected no new loci among SGIT loci. The clumping of UGITs revealed 379 loci, of which 246 were new. At the same time, the clumping of only original traits allowed 574 loci to be detected, of which 187 could not be detected by analyzing SGIT or UGITs. Thus, the joint analysis of SGIT and UGITs increased the number of associated loci by more than 42.8%. Figure 4C reflects the overlapping between significantly associated loci for 11 analyzed traits. There is a weak albeit non-zero overlap between loci for UGITs and SGIT, although the genetic correlation between them is zero. It could be due to the conservative settings of the clumping procedure, which tends to clump together closely located loci, and is due to some level of unspecificity of the SHAHER. 

Next, we checked how enriched gene sets overlap between SGIT, UGITs and the original traits (see Figure 4D). Significant results (FDR < 5%) of enriched gene sets and tissue enrichment analyses are presented in Appendix A. As expected, the heatmap of the overlapping gene sets looks similar to the heatmap of genetic correlations and the heatmap of the overlapping loci. Moreover, there was almost no overlap between SGIT and any UGIT. For the original traits, the number of enriched gene sets varied a lot: from four for the waist to 825 for the hip circumference. For BMI UGIT, the number of enriched gene sets was 1608, which was almost ten times the value for BMI (192). 

Finally, we obtained GIP1 GWAS statistics and calculated the genetic correlations between SGIT and GIP1. The genetic correlation was higher than 0.97. 

## 4. Discussion

We developed a new fast and efficient framework which allows us to decompose the heritability of each trait from a given set of traits into two components. One of them is explained by shared genetic factors common to all traits. Another one is explained by unshared genetic factors specific for each trait. The framework not only decomposes heritability but also identifies SNPs associated with the shared and unshared genetic effects. To our knowledge, this framework is unparalleled. It has an additional advantage: it uses GWAS summary statistics obtained for original traits and does not require raw genotype or phenotype data.

We compared the performances of MaxSH and GIP in identifying the shared genetic components. GIP calculates the linear combination coefficients via the eigenvalues of the genetic covariance matrix and can be considered a close approximation to MaxSH. In our simulations, GIP and MaxSH were similar in almost all scenarios, with MaxSH being somewhat superior in terms of the power (total heritability) and quality (shared heritability). If obtaining genetically independent phenotypes is not the aim, we suggest using SHAHER, because it is more robust and gives additional metrics like SGI contributions to the heritability of the original traits. 

The framework is computationally effective. The stage using sumCOT is the most time consuming. However, it only takes several minutes for an average computer to conduct a GWAS of a linear combination of traits with 6M SNPs using a C++ implementation of the sumCOT. MaxSH, based on numerical optimization procedures, and the other parts of the framework take seconds. 

The proposed sumCOT method can be applied as an independent tool to address additional tasks. One of them is making a summary-level adjustment of traits by other traits using the same scheme as was used for obtaining the UGIT GWAS statistics. This can be helpful, for example, for ridding the studied trait’s genetic component of the genetic component that was caused by the confounding or unaccounted effects of assortative mating or family effects, which is quite a problem in GWAS at the biobank scale [15,23]. Another task is a GWAS for the trait that appears as a linear combination of the original traits. The sumCOT method is robust to differences in sample sizes used for GWASs of original traits and is applicable to different GWAS models (Cox, linear or logistic). 

The main interest in the application of the SHAHER framework lies in the possibility of obtaining novel biological insights into a trait’s heritability composition. This can be achieved by the application of a huge variety of in-silico follow-up techniques to SGIT and UGITs. SGIT is of interest by itself, but we also emphasize the importance of the comparison of shared and unshared effects for each trait. In our real data application, the most remarkable case is BMI in the set of anthropometric traits (see Figure 4C). We found 246 and 1608 significantly enriched gene sets for SGIT and the UGIT of BMI, respectively, with negligible overlapping between them of size 56. By analyzing BMI only, we would have detected only 192 enriched gene sets. By analyzing each of the impacts separately, we dramatically increased the number of observed unique gene sets (1798 in total for both SGI and UGI). This means that each sub-phenotype controlled by SGF and UGF is less heterogeneous than the original trait. According to the significant gene sets, UGIT of BMI (see Appendix A) controls some structural changes in body compositions and bone formation, while SGIT is involved in some general signaling pathways and pathways related to nervous system development and probably to general psycho-social aspects of BMI, obesity and other anthropometric traits [24]. Note that all new loci were associated with UGITs. We can speculate that these new SNPs were detected due to the decreased genetic heterogeneity of UGITs compared to the original traits. 

To validate the findings of SHAHER, we have compared the association results for anthropometric and psychometric traits with the biggest publicly available GWAS results for BMI and MDD. The idea is as follows: if the locus was not significant on the original trait but was detected on SGIT or corresponding UGIT, it will be detected on the original trait if the GWAS sample size is increased. For BMI we have used the largest meta-analysis of the UK Biobank and GIANT GWAS (*N* = 806,834) to date [25] (see Appendix A). Among 264 loci significant on SGIT and BMI UGIT but not on BMI, 57 loci (22%) became significantly associated with BMI in the biggest GWAS. If we consider only loci associated with SGIT, the validation ratio is higher: out of 86 loci, 49 (57%) were significant in the biggest GWAS. For MDD we have used the biggest meta-analysis of the UK Biobank and PGC GWAS (*N* = 500,199) to date [26] (see Appendix A). Among 58 loci significantly associated with SGIT and MDD UGIT but not with MDD, seven (12%) became significant in the biggest MDD GWAS. The similar validation of lipid loci was not performed since there was no bigger GWAS available in the open access. The validation of the loci on the biggest GWAS is not a proper replication, however it still greatly increases the confidence that detected by SHAHER loci are true positives. 

Although SHAHER is effective, it has several limitations. First, when trait-trait genetic correlations are weak, it is expected that the contributions of these traits to the shared heritability will be small as well. In this case, MaxSH may overestimate these contributions. Secondly, the framework is applicable only if the number of traits is no less than three. In the case of three traits, the performance is limited and the SHAHER results should be interpreted with caution. We have shown in simulations and real dataset examples that MaxSH works better at higher numbers of genetically correlated traits being analyzed. However, an increase in the number of weakly correlated traits leads to a decrease in the proportion of SNPs associated with all traits simultaneously and to a decrease in the efficiency of the framework. Thirdly, although the set of SNPs identified by the SGIT GWAS is enriched for the SGF, each SNP should be interpreted with caution for whether it is shared or not, because SHAHER has some level of nonspecificity. Finally, if any confounding effects were included in the GWAS of the original traits, these effects are amplified in the SGIT [15]. The confounding effects can be controlled easily using special methods like LD score regression [16], although this method fails to distinguish a polygenic component if the trait was measured in the sample with the assortative mating or family effects. Thus, we suggest a thorough check of the original GWAS for the presence of any effects of possible confounders before proceeding to SHAHER. In principle, if the LD score regression intercept was estimated, it is possible to correct for residual inflation by adjusting the standard errors of the effects by multiplying them by the square root of the intercept. 

We should highlight the distinctive specificity of SHAHER which distinguishes it from existing approaches for multivariate analysis. There are a lot of frameworks that allow for the incorporation of several correlated traits in one analysis to increase the power of mapping [8,9,10,11,12,13]. Our framework is not aimed to increase the power of mapping itself (although empirically we showed that SHAHER has higher power compared to univariate analyses). Our framework is aimed to estimate the shared and unshared heritability and to identify the shared and unshared genetic factors. Therefore, we did not compare the power of SHAHER with the powers of existing approaches and do not expect it to have the highest power among them. Moreover, our definition of shared genetic factors is stricter than just the pleotropicity of all analyzed traits. This is why using multivariate approaches aimed to increase the power of mapping is not the optimal way to identify shared genetic factors. 

In conclusion, we propose a novel effective framework for analysis of the shared genetic background for a set of genetically correlated traits using GWAS summary statistics. The framework allows us to obtain novel biological insights into the trait’s genetic impact composition. By analyzing shared and unshared genetic impacts separately, we increased the number of identified loci and observed unique gene sets, identified genetic mechanisms that are common for all traits or specific for every trait. Of note, sumCOT can be used as a stand-alone method for obtaining GWAS results of the linear combination of the traits using their summary statistics.

## Figures and Tables

**Figure 1 genes-13-01694-f001:**
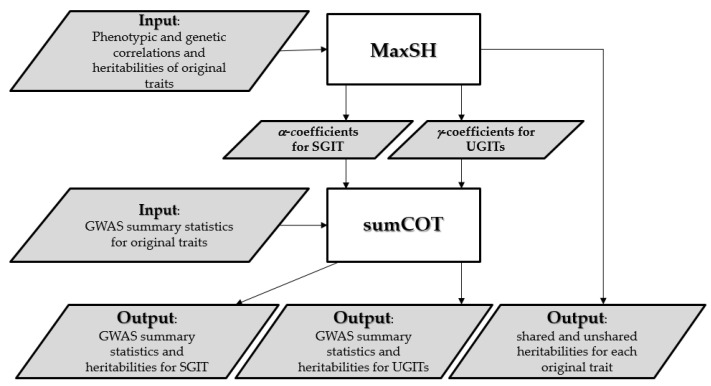
Flowchart of the SHAHER framework. Details are given in the text.

**Figure 2 genes-13-01694-f002:**
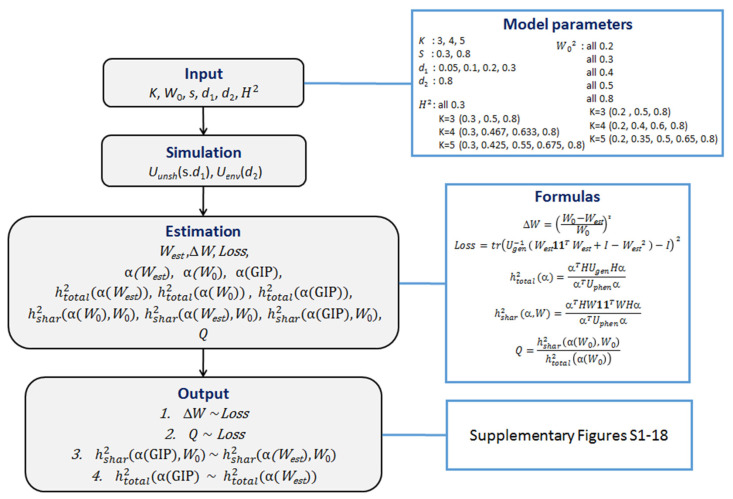
A schematic depicting the overall workflow of a simulation study. All details are given in the text.

**Figure 3 genes-13-01694-f003:**
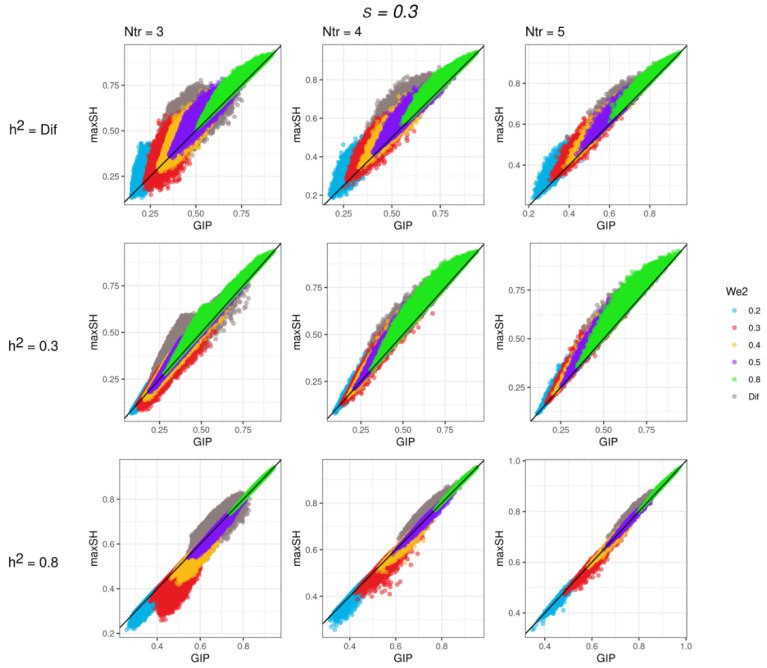
The plot of the shared heritability of SGIT (the maxSH method) versus the shared heritability of GIP1 (the GIP method). For different *h*^2^ and different number of traits (Ntr) for *s* = 0.3.

**Figure 4 genes-13-01694-f004:**
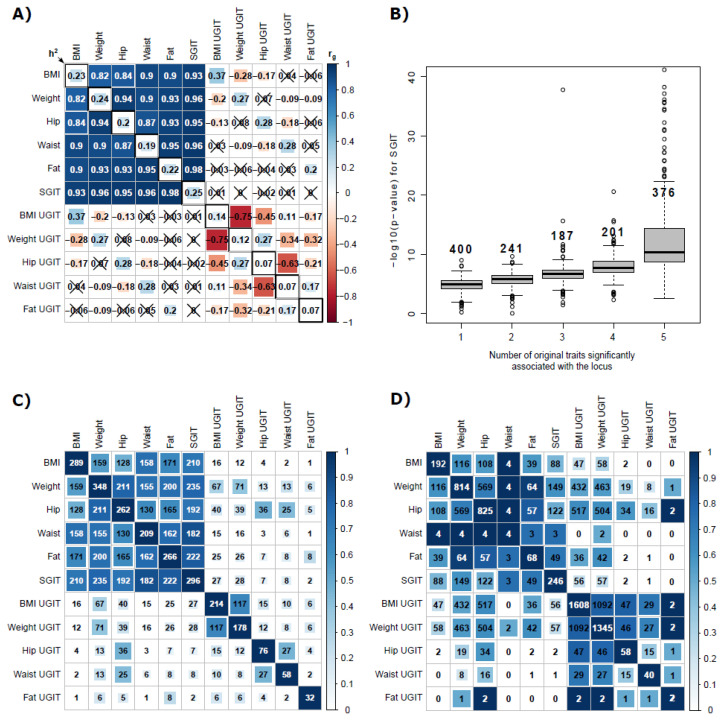
Results of the application of SHAHER to anthropometric traits. (**A**) The heatmap of genetic correlations between the original, SGI and UGI traits. The number, color strength and size of the squares in the matrix show the values of the correlation coefficients between the traits. The diagonal elements represent heritabilities. Crossed out values indicate insignificant correlations. (**B**) Boxplots of −log_10_(*p*-value) for the SGIT with respect to the number of the original traits significantly associated with the locus. Two outliers for loci with −log_10_(*p*-value) > 40 are omitted. The number at the top of the boxplot corresponds to the number of significant SNPs. (**C**) The heatmap of the numbers of overlapping loci between traits. The numbers in the cells represent the absolute numbers of overlapping loci. The color strength and size of the squares in the cells show the relative scaled number of overlapping loci (on a scale from 0 to 1). The diagonal elements represent the number of loci found for every trait. (**D**) The heatmap of the numbers of overlapping gene sets between traits. The color strength and size of the squares in the cells show the relative scaled number of overlapping gene sets (on a scale from 0 to 1). The diagonal elements represent the number of gene sets found for every trait.

**Table 1 genes-13-01694-t001:** Number of significant loci (*p*-values < 5 × 10^−8^) identified for each trait applying SHAHER for three data sets.

Trait Name	Number of Significant Loci		
	Real Trait	SGIT *	UGIT *
**Anthropometric Traits**			
**BMI**	289	296 (210)	214 (16)
**Weight**	348	296 (235)	178 (71)
**Hip**	262	296 (192)	76 (36)
**Waist**	209	296 (182)	58 (6)
**Fat**	266	296 (222)	32 (8)
**Psychometric Traits**			
**BIP**	12	57 (8)	2 (0)
**MDD**	3	57 (0)	2 (1)
**SCZ**	92	57 (26)	2 (0)
**Happiness**	0	57 (0)	1 (0)
**Lipid Traits**			
**LDL**	85	97 (69)	43 (31)
**Triglycerides**	71	97 (26)	59 (30)
**Cholesterol**	101	97 (84)	51 (21)

* number of loci overlapping with those identified using the original trait is given in parentheses.

## Data Availability

UK Biobank GWAS summary statistics for anthropometric traits and for subjective well-being: http://www.nealelab.is/blog/2017/7/19/rapid-gwas-of-thousands-of-phenotypes-for-337000-samples-in-the-uk-biobank, accessed on 1 September 2020. GWAS summary statistics for psychometric trait: the Psychiatric Genomics Consortium, https://www.med.unc.edu/pgc/download-results/, accessed date September 2020. GWAS for lipid traits: the Global Lipid Genetics Consortium, http://csg.sph.umich.edu/willer/public/lipids2013/, accessed on 1 September 2020. The biggest to date MDD GWAS meta-analysis of the UK Biobank and PGC: https://datashare.ed.ac.uk/handle/10283/3203, accessed on 1 August 2022. The biggest to date BMI GWAS meta-analysis of the UK Biobank and GIANT: https://portals.broadinstitute.org/collaboration/giant/index.php/GIANT_consortium_data_files, accessed on 1 August 2022.

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
