# Peer review of "A Novel Framework for Analysis of the Shared Genetic Background of Correlated Traits"

_genes, 2022, doi:10.3390/genes13101694_

Round 1

Reviewer 1 Report

This is an interesting study of development of a novel framework for analysis of the shared genetic background of correlated traits. The study described detailed algorithm development, and then applied the model to different simulated scenarios, as well as three sets/types of real data to test the framework. However, the presentation of the results was quite weak; and the manuscript was somehow disorganized. Some comments:

1. Page 2 "Abbreviations and terms" have been described in the previous section; and it's unnecessary to occupy a major section of the main text body;

2. Page 3-6: Most of the texts would be better to move to the Methods part;

3. There should be much more main simulation and real data analysis results presented in the Results section, e.g., Simulation results: there were 18 figures about the simulation results under different scenarios. 1-2 selective or simplified simulation results should be presented in the main Results part. Equally, 1-2 selected and/or summarized figures/tables from each real data set could be presented in the main Results part, such as anthropometric traits, PGC, and Lipid concentrations, respectively.

4. The quality and resolution of the figures could be improved for better visual effect.

5. The discussion part could be further expanded to compare the results of the current study with some similar cross-disorder studies to further demonstrate the validity and utility of such approach and program.       

Author Response

Dear Editor,

We would like to thank you and Reviewers for helpful comments and suggestions. Please find attached our point-by-point answers below. We revised our manuscript accordingly. We hope that you and the reviewers will find the revised manuscript suitable for publication in Genes.

Please note that we have found and corrected minor mistakes in figures related to real data analysis. The changes do not alter the conclusions. 

Yours Sincerely,

also on behalf of other authors,

  Dr. Yakov Tsepilov

Reviewer 1.

  1. Page 2 "Abbreviations and terms" have been described in the previous section; and it's unnecessary to occupy a major section of the main text body;

A.: “”Abbreviations and terms” have been omitted.

  1. Page 3-6: Most of the texts would be better to move to the Methods part;

A.: We have changed the structure of the manuscript and placed the description of Materials and Methods before the Results. We moved the description of the framework in the Methods section.

  1. There should be much more main simulation and real data analysis results presented in the Results section, e.g., Simulation results: there were 18 figures about the simulation results under different scenarios. 1-2 selective or simplified simulation results should be presented in the main Results part. Equally, 1-2 selected and/or summarized figures/tables from each real data set could be presented in the main Results part, such as anthropometric traits, PGC, and Lipid concentrations, respectively.

A.: We have revised the Results section and added one additional figure (selection of the figures from the simulations) and one additional table (summarizing the results from real data analysis) to the main text. The text and supplementary results were modified accordingly.

  1. The quality and resolution of the figures could be improved for better visual effect.

A.: All figures from the main text are attached to the submission in high quality as separate files.

  1. The discussion part could be further expanded to compare the results of the current study with some similar cross-disorder studies to further demonstrate the validity and utility of such approach and program.

A.: We have expanded the Discussion by validation of the loci on newer bigger GWASs for BMI and MDD. We added the following paragraph: “To further validate the findings of SHAHER, we have compared the association results for anthropometric and psychometric traits with the biggest publically available GWAS results for BMI and MDD. The idea is following: if the locus was not significant on original trait but was detected on SGIT or corresponding UGIT – it will be detected on the original trait if the GWAS sample size is increased. For BMI we have used the biggest to date meta-analysis of the UK Biobank and GIANT GWAS (N = 806,834) [26] (see Supplementary Table 3a). Among 264 loci, significant on SGIT and BMI UGIT but not on BMI, 57 loci (22%) became significantly associated with BMI in the biggest GWAS. If we consider only loci, associated with SGIT, the validation ratio is higher – out of 86 loci, 49 (57%) were significant in the biggest GWAS. For MDD we have used the biggest to date meta-analysis of the UK Biobank and PGC GWAS (N = 500,199) [27] (see Supplementary Table 3b). Among 58 loci significantly associated with SGIT and MDD UGIT but not with MDD, 7 (12%) became significant in the biggest MDD GWAS. The similar validation of lipid loci was not performed since there was no bigger GWAS available in the open access. The validation of the loci on the biggest GWAS is not a proper replication, however it still greatly increases the confidence that detected by SHAHER loci are true positives.”

Reviewer 2.

The current manuscript describes a novel effective framework for analysis of the shared genetic background for a set of genetically correlated traits using SNP-level GWAS summary statistics. I believe this new framework will be useful to some scientists working on the GWAS. The methods and the results were deliberately explained in the manuscript, and the results were discussed with relevant literature.

A.: Thank you! We have modified the structure of the text according to the comments of the first Reviewer for better readability.

Reviewer 2 Report

The current manuscript describes a novel effective framework for analysis of the shared genetic background for a set of genetically correlated traits using SNP-level GWAS summary statistics. I believe this new framework will be useful to some scientists working on the GWAS. The methods and the results were deliberately explained in the manuscript, and the results were discussed with relevant literature. 

Author Response

(The authors gave the same response as above.)

Round 2

Reviewer 1 Report

The revised manuscript has been improved significantly; and has addressed all my comments and concerns. One more minor suggestion: data and code availability should be listed after the main text. In addition to the code availability, the authors should also provide the links/sources of data sets described and used in the manuscript. 

Author Response

We have added "Code Availability" and "Data Availability" after the main text.